# Comparative Analysis of Health- and Vision-Related Quality of Life Measures among Trinidadians with Low Vision and Normal Vision—A Cross-Sectional Matched Sample Study

**DOI:** 10.3390/ijerph20146436

**Published:** 2023-07-24

**Authors:** Kingsley K. Ekemiri, Edith N. Botchway, Ngozika E. Ezinne, Nikolai Sirju, Tea Persad, Hlabje Carel Masemola, Sherphard Chidarikire, Chioma C. Ekemiri, Uchechukwu Levi Osuagwu

**Affiliations:** 1Optometry Unit, Department of Clinical Surgical Sciences, Faculty of Medical Sciences, University of the West Indies, St Augustine Campus, St. Augustine 685509, Trinidad and Tobago; 2Department of Optometry, College of Health Sciences, University of Kwazulu-Natal, Westville Campus, Durban 3629, South Africa; 3Brain and Mind Group, Clinical Sciences, Murdoch Children’s Research Institute, Parkville, VIC 3052, Australia; 4Department of Pediatrics, University of Melbourne, Parkville, VIC 3010, Australia; 5Department of Optometry, Faculty of Health Sciences, University of Free State, Bloemfontein 9301, South Africa; 6School of Nursing and Midwifery, College of Health, Medicine and Wellbeing, University of Newcastle, Callaghan, NSW 2308, Australia; 7Department of Health Promotion, The University of the West Indies, St. Augustine Campus, St. Augustine 685509, Trinidad and Tobago; 8Bathurst Rural Clinical School, School of Medicine, Western Sydney University, Bathurst, NSW 2795, Australia

**Keywords:** vision impairment, low vision, eye diseases, quality of life, Trinidad

## Abstract

This cross-sectional study investigated the health-related and vision-related quality of life measures of adults with low vision compared to healthy individuals in Trinidad and Tobago. The health-related quality of life (HRQOL-14) and the National Eye Institute Visual Functioning Questionnaire (NEI-VFQ-25) were administered to 20 participants with low vision caused by diabetic retinopathy, retinitis pigmentosa, glaucoma, and macular degeneration, as well as 20 participants with no visual problems (control). Participants were recruited from the University Eye Clinic in Trinidad and Tobago. Compared to the controls, more participants in the low-vision group had lower age-adjusted NEI-VFQ-25 scores (48.3% vs. 95.1%; *p* < 0.001), had poor general (47.5% vs. 10%, *p* = 0.004) and mental (100% vs. 10%, *p* < 0.042) health, experienced greater activity limitation due to impairment or health problems (85% vs. 20%, *p* < 0.001), needed help with personal care (27.5% vs. 0%, *p* < 0.009) and daily routine (67.5% vs. 0%, *p* < 0.001), and experienced sleep problems (97.5% vs. 65%, *p* < 0.001) and symptoms of anxiety (100% vs. 90%, *p* = 0.042). All the diabetic retinopathy participants (100%, *p* = 0.028) had two or more impairments or vision problems compared to none in the other low-vision participants. In summary, the HRQOL-14 and NEI-VFQ-25 scores were significantly reduced in low-vision participants, who also demonstrated a greater vulnerability to poor quality of life in the presence of diabetes retinopathy. These findings have important clinical implications regarding offering appropriate support and interventions to improve quality of life outcomes in individuals with low vision.

## 1. Introduction

Low vision rehabilitation services have evolved over the years and require eye health professionals to work within a multidisciplinary team across health, social/community sector, and non-governmental organizations to improve the effectiveness and accessibility of low vision care [1]. The World Health Organization (WHO) defines low vision as an impairment of visual functioning with a distance visual acuity (VA) of less than 6/18 or visual field of equal to or less than 20 degrees in the better eye, even after refractive correction and medical or surgical intervention [2]. Vision loss has many causes, including age-related macular degeneration and glaucoma; requires improvements in preventive and rehabilitative interventions; and often impacts functional well-being and quality of life (QoL) [3,4], which further deteriorates with increasing severity of VI [5].

According to the world report on vision, there are at least 2.2 billion people around the world with visual impairment (VI), of which at least 1 billion are preventable [6]. It is estimated that blindness and vision loss resulted in 26.5 million global years of healthy life lost due to disability in 2019 and 3.1% of total global years of healthy life lost due to disability [1]. Data from 2020 show that cataracts, glaucoma, under-corrected refractive error, age-related macular degeneration, and diabetic retinopathy are the leading causes of blindness globally in those aged 50 years and older, while leading causes of moderate to severe VI were under-corrected refractive error and cataracts [5].

In the Caribbean, a prevalence of 1.7% and 5.7% were recorded for blindness and low vision, respectively [7]. It is difficult to make direct comparisons of prevalence rates in this region since data collection methods, definitions of VI, and population characteristics vary across the region. For instance, 3.8% of Jamaicans were reported to have moderate and severe VI in one study [8], while a prevalence rate of 2.8% was reported for VI (moderate and severe) resulting predominantly from cataracts, glaucoma, uncorrected refractive error, diabetic retinopathy, and macular degeneration in Trinidad and Tobago (International Agency for the Prevention of Blindness), both of which are lower than the global VI average of 4.5% [9]. In Trinidad and Tobago, glaucoma and cataracts were also recorded as the major causes of blindness while uncorrected refractive error and cataracts were the major causes of low vision, respectively [1,10]. These studies indicate the pervasive nature of low vision and highlight the need to understand health outcomes (e.g., QoL), especially in vulnerable individuals living with these common vision problems.

The Lancet Global Health Commission on global eye health suggests that several Sustainable Development Goals (SDGs) can be achieved through adequate eye health [1]. Equitable access to and achievement in education and the workplace are hampered by impaired vision and poor eye health, which have a harmful effect on health-related quality of life (HRQOL-14) [1]. The World Report on Vision proposes integrated people-centered eye care (IPEC) and seeks to encourage action in countries to address these challenges [4]. This IPEC approach to health system strengthening aims to shape an underpinning for service delivery to address population needs. “IPEC refers to eye care services that are managed and delivered to assure a continuum of promotive, preventive, treatment and rehabilitative interventions to address the full spectrum of eye conditions, coordinated across the different levels and sites of care within and beyond the health sector, and according to people’s needs throughout their life course” [4].

The impacts of VI and blindness are wide-reaching, including an increased risk of falls, cognitive impairment and dementia, depression, disability, loss of independence, and poor QoL [3,4]. Poor visual function has been associated with poor mental health and poor health-related (HRQoL) and vision-related quality of life (VR-QoL) in adults [11,12]. In the only study from Trinidad and Tobago (T&T) on the impact of vision loss on QoL, Braithwaite and colleagues used data from a population-based national cross-sectional survey to show an independent association between less severe categories of distance and near VI (NVI) and quality-adjusted life lost (QALYS) to vision impairment in adult participants aged 40 years and above [13]. In an observational study on 128 persons attending a rehabilitation center for visually impaired adults in the Netherlands [14], the authors reported that QoL was more reduced among visually impaired adults than those with other chronic conditions, including type 2 diabetes, coronary syndrome, and hearing impairments, but less than those with stroke, multiple sclerosis, chronic fatigue syndrome, major depressive disorder, and severe mental illness. The deterioration in visual function has a significant impact on performing daily functions and leisure activities, leading to impaired efficiency and compromised independence of an individual [10]. Some studies have also suggested that VI and blindness are associated with an increased risk of mortality [9]. Vision loss can be frightening and overwhelming not only to the individuals experiencing it, but also to their families, friends, and society. Thus, vision loss is a largely preventable global problem, often affecting vulnerable communities with limited resources to prevent or manage this problem. 

VI is a pervasive problem in T&T [11], and such impairments have been associated with low QoL in several studies around the world [13,14,15,16]. Access to eye care and affordability are persistently significant issues in T&T, despite the availability of numerous effective health care interventions for the prevention and control of primary eye conditions, potentially affecting the QoL of people with VI in this country [17]. To date, however, no observational study has investigated the relationship between VI and QoL in adults in T&T, with the only published evidence coming from a national eye health survey [10] that was not specifically designed for this purpose. Such specific investigations are needed to inform clinicians, researchers, and policymakers about the impact of low vision on the livelihood of affected individuals. This can lead to the development of further research, as well as evidence-based tailored interventions and social support services aimed at improving the functional well-being and QoL of individuals with low vision in T&T. 

T&T is a high-income country with a well-managed public debt, solid human capital, sufficient financial safeguards, and political stability. The public spending of Trinidad and Tobago on health is 3.7 percent of its Gross Domestic Product (GDP), which is lower than their regional average of 3.8 percent [12]. Investigating the HRQOL-14 and NEI-VFQ-25 of low-vision patients will provide an overview of the impact of the VI on the patient’s life from their perspective. The current study aims to investigate the following: (1) the differences in HRQoL and VR-QoL outcomes between adults with low vision and healthy adults in Trinidad and Tobago and (2) whether the HRQoL and VR-QoL outcomes differ within the low-vision group based on their ocular conditions (i.e., glaucoma, retinitis pigmentosa, diabetic retinopathy, and age-related macular degeneration). Based on current research showing poor QoL in low-vision patients from other regions around the world, we hypothesize that adults with low vision in T&T will present with lower HRQoL and VR-QoL compared to controls. Evidence from this study is expected to provide an overview of the impact of the VI on a patient’s QoL in T&T.

## 2. Materials and Methods

### 2.1. Setting and Design

This was a cross-sectional study aimed at assessing HRQoL and VRQoL of adults with low vision compared with those without low vision in T&T. All participants were recruited from the University of the West Indies (UWI) optometry clinic at Couva Training Facility, located in the western-central part of Trinidad. 

### 2.2. Participants and Eligibility Criteria 

The participants were 20 adults with low vision and 20 adults with normal vision (controls). The low-vision group included adults aged 18 years and over who had previously attended the UWI Optometry Clinic (between January and May 2022), had a formal diagnosis of low vision, and were already using prescribed low vision aids. Causes of low vision in this group included glaucoma, retinitis pigmentosa, diabetic retinopathy, or age-related macular degeneration. The control group was made up of adult participants (staff and students) from the university community without diabetic retinopathy, retinitis pigmentosa, glaucoma, and macular degeneration, who had a best corrected visual acuity of 20/20. This group underwent comprehensive eye tests (e.g., visual acuity, intraocular pressures, retinal examination, and slit lamp examination) to rule out any ocular problems. 

Exclusion criteria for both groups (i.e., low-vision and controls) were (1) individuals under 18 years old, (2) individuals with corneal defects or anterior surface eye diseases such as keratoconus, Fuchs’ endothelial dystrophy, and bullous keratopathy, and (3) individuals with cognitive impairments such as Alzheimer’s or dementia recorded in their hospital records.

### 2.3. Procedures

Patients who visited the UWI Optometry Clinic during the study period and who met the eligibility criteria to participate in the study were consecutively selected using a convenient sampling method and allocated to the respective groups (low-vision and control). They were invited to participate in the study through an information pamphlet which was distributed during clinic days. We obtained written and verbal consent from all participants, and only those who consented were enrolled in the study. Before data collection, information regarding the study was provided to all participants via a participant information sheet and consent form. Appointments were scheduled for a telephone interview with each participant where the CDC HRQOL-14 and NEI-VFQ-25 questionnaires were completed. All interviews were conducted by a single researcher (KE), who received extensive training on how to administer the questionnaires over the phone with low-vision patients. It took approximately 30 min to complete the questionnaire over the phone per participant. When a participant was unable to complete the questionnaire within the period of the appointment, another appointment was scheduled to complete the questionnaire.

### 2.4. Measures

Health-related quality of life. The US Center for Disease Control and Prevention Health-Related Quality of Life (CDC HRQOL-14) [14] was used to rate participants’ perceived overall health status and the influence of various symptoms on their daily activities. It comprised a healthy day core module (4 items), activity limitations module (5 items), and a healthy day symptom module (5 items). The first question on the healthy days module rated overall health on a five-point Likert-like scale (0 to 5, Excellent to Poor). We report a binarized version of this variable (1 = Excellent to Good, 2 = Fair to Poor). Questions 2–3 assessed the number of days the respondent has had poor physical and mental health, and question 4 evaluated the number of days during which poor physical/mental health prevented the respondent from doing their usual activities (all on a scale of 0 to 30). Questions 1, 4, and 5 on the activities limitation module assessed various aspects of activity limitation on a ‘Yes’ or ‘No’ response scale. Item 2 evaluated the kinds of impairments that limit the respondent’s activities on a scale of 0 (None) to 14 (Other impairment/problem). We recorded this variable to group all participants endorsing more than one impairment together while presenting details on the proportion of participants who endorsed a single impairment. Item 3 assessed how long the respondent’s usual activities have been limited due to the selected impairment(s)/health problem(s), coded as 1 = Days, 2 = Weeks, 3 = Months, and 4 = Years. Questions 1 to 4 on the healthy days symptom module rated the presence of specific symptoms that may have led to activity limitation in the last 30 days, including pain, sadness, anxiety, depression, and lack of sleep. Question 5 assessed the number of days in the last 30 days that the respondent felt very healthy/full of energy. With all the questions assessing outcomes within the 30-day timeframe, a lower number of unhealthy days represented higher HRQoL, and we reported mean scores for these items in line with reporting formats used in some published studies [5,18]. The CDC HRQoL-14 is a valid, reliable tool for use in the adult population, and on testing, it demonstrated satisfactory validity (average Cronbach’s α coefficient score of 0.76).

Vision-related quality of life. The validated short 25-item National Eye Institute Visual Functioning Questionnaire (NEI-VFQ-25) was used to assess patient-reported NEI-VFQ-25 [15]. Participants responded to the questions based on their binocular visual function at its best, whether aided or unaided. The questionnaire determines the individual’s overall visual status and their quality of life based on their responses to the questions. The use of the short version of the questionnaire was preferred to the original 50-item questionnaire because survey length can negatively impact the quality of data [15]. The 25-item version produces 12 subscales that are grouped into general health (non-vision-specific items) and vision-specific subscales, including questions on general health and vision status, ocular pain, distant and near vision, ability to function socially, mental health, difficulty carrying out roles, dependency on others, driving, color vision, and peripheral vision. We reported the 12 subscale scores and composite score of this measure, ranging from 0 to 100, with higher values indicating better NEI-VFQ-25 [16]. The items on color vision and peripheral vision were recorded from a four-point scale to a two-point scale (extreme/moderate difficulty and little/no difficulty). The NEI-VFQ-25 questionnaire was tested for its internal reliability and validity in this sample population, with a Cronbach’s α coefficient average score of 0.83 demonstrating an overall good reliability.

### 2.5. Statistical Analysis

All analyses were performed using SPSS Version 29 (IBM Corp, Armond, Australia). Data were checked for compliance with relevant statistical assumptions using descriptive statistics and frequencies. Missing values were reported in the footnotes of tables, where applicable. Independent sample *t*-test, analysis of variance (ANOVA), and χ^2^ tests were used to assess demographic differences in age and sex between the low-vision and control groups, as well as within the low-vision group (i.e., glaucoma, retinitis pigmentosa, diabetic retinopathy, and age-related macular degeneration). 

To address aim one, differences in HRQOL between the low-vision and control groups were assessed using Generalized Linear Models (Linear, Binary Logistic, Poisson, and Ordinal Logistic), since these outcome variables were not normally distributed. Separate analyses were conducted for each item on the CDC-HRQoL-14. Similar models were used to evaluate group differences on VRQoL, using scores on the NEI-VFQ-25. We present results for both unadjusted and adjusted models, with the latter controlling for group differences in age. 

For aim two, χ^2^ tests and one-way ANOVA models were used to evaluate differences in HRQoL and VRQoL outcomes among the ocular condition groups (i.e., glaucoma, retinitis pigmentosa, diabetic retinopathy, and age-related macular degeneration). Effect size estimations for χ^2^ tests were based on Cramer’s V values, interpreted as mild (≤0.2), moderate (0.3–0.6), and severe (>0.6). Effect size estimations for the ANOVA models were based on eta squared values, interpreted as small effect (η^2^ = 0.01), medium effect (η^2^ = 0.06), and large effect (η^2^ = 0.14). A value of *p* < 0.05 was considered statistically significant for all analyses.

## 3. Results

### 3.1. Demographic Characteristics of the Sample

The process of selection and the distribution of the low-vision participants by disease status have been presented in Figure 1. The majority of the low-vision group had retinitis pigmentosa (n = 9). There were no significant between-group differences in age (*p* = 0.063) and sex (*p* = 0.568). An evaluation of age and sex differences between the low-vision and control groups showed a statistically significant age difference (*p* < 0.001), with the low-vision group reporting a higher average age (mean = 49.3, SD = 12.5) compared to the control group (mean = 29.0, SD = 9.8). The two groups, including the low-vision group (males 45.0%, females 55.0%) versus the control group (males 35.0%, females 65.0%), were comparable with regards to sex (*p* = 0.519). Hence, age was statistically adjusted in subsequent contrasts between the low-vision and the control group. 

### 3.2. Differences in HRQoL between the Low-Vision and Control Groups

Table 1 presents results for generalized linear models assessing the differences in HRQoL outcomes between the low-vision and control groups, including unadjusted models and adjusted models, controlling for age.

Healthy Days Core Module. Both unadjusted and adjusted models showed no statistically significant differences between the low-vision and control participants for poor general health and for the impact of mental and physical health on activity limitation (*p* > 0.05). Unadjusted models showed statistically significant differences between the low-vision and control groups for a number of days for which they experienced poor physical health (*p* = 0.027, higher number of days in low-vision group) and mental health (*p* = 0.012, higher number of days in the control group). However, no statistically significant differences were found between the groups on these outcomes after controlling for the effect of age.

Activities Limitation Module. The proportion of participants who indicated that their activities were limited due to impairment was significantly higher in the low-vision group compared to controls in both unadjusted (*p* < 0.001) and adjusted (*p* = 0.006) models. The main problems/impairments reported in the low-vision and control groups were significantly different in both unadjusted (*p* < 0.001) and adjusted models (*p* < 0.005). Chi-squared tests showed that all participants who endorsed no problems (60%) and who indicated that their activities were limited by back/neck problems (5%), heart problems (5%), and emotional problems (20%) were in the control group. All participants who indicated that eye/vision problems limited their activity levels were in the low-vision group (45%), and this group also had a greater proportion of participants endorsing more than one problem (55%) compared to the control group (10%). With regards to the duration of activity limitation, no significant difference was found between the two groups in both unadjusted and adjusted models, although more low-vision participants endorsed being affected for years compared to control participants. Likewise, responses from both groups were similar on whether they needed support with personal care or routines (both *p* > 0.05).

Healthy days symptom module. Generalized linear models (unadjusted and adjusted) examining the number of days in the last 30 days where participants experienced pain, moodiness, or felt fully energetic did not show significant differences between the low-vision and control participants (all *p* > 0.05). However, we found a significantly higher number of anxious/worrying days in controls compared to the low-vision participants in both unadjusted (*p* = 0.005) and adjusted (*p* = 0.017) models. The control group also reported a higher number of days of sleep problems compared to the low-vision group in the unadjusted model (*p* = 0.042), but this difference disappeared after adjusting for age (*p* = 0.109). 

### 3.3. Differences in VR-QoL between the Low-Vision and Control Groups

Table 2 presents the results of the unadjusted and adjusted analyses for the differences in the mean NEI-VFQ-25 scores between the low-vision group and control groups. Except for the color vision models, results showed statistically significant differences between the two groups for all unadjusted and adjusted models (all *p* < 0.005). For all those subscales of the NEI-VFQ-25, the low-vision group reported significantly poorer VRQoL compared to the control group, and age was not a significant predictor in any of the models. No control participants reported extreme/moderate color vison or peripheral vision problems, while 20% and 73.7% of the low-vision group reported these problems, respectively. 

### 3.4. HRQoL Differences among the Ocular Condition Groups

As indicated in Table 3, the four ocular condition groups presented comparable HRQoL outcomes on items of the CDC-HRQoL-14 (all *p* > 0.05). Results for the healthy days core module and healthy days symptom module merged with small to large effect sizes, while the results for the activities limitation module were associated with small to medium effect sizes. 

### 3.5. VRQoL Differences among the Ocular Condition Groups

Figure 2a–m presents results from ANOVA models evaluating the effect of ocular conditions on VRQoL. Although the glaucoma group showed a trend towards poorer VRQoL on several scales, no statistically significant difference was found among the four ocular condition groups (all *p* > 0.05). Effect sizes ranged between small (near activities, distance activities, and peripheral vision scales), medium (general vision, ocular pain, social function, mental function, color vision scales, and the VRQoL composite score), and large (general health, role difficulties, dependency, and driving scales).

## 4. Discussion

The study investigated effects of low vision on HRQOL and VRQoL by comparing outcomes in adults with low vision and healthy controls. Compared to the control group, participants with low vision reported poorer HRQoL (particularly related to activity limitation) and VRQoL after controlling for age. However, HRQoL and VRQoL outcomes did not differ among the ocular condition groups. These findings highlight greater vulnerability to poor quality of life in individuals with low vision and indicate the need for interventions and support services aimed at improving outcomes in this domain. 

In support of our hypothesis, the low-vision group reported poorer HRQoL on some items of the CDC HRQoL-14 compared to controls. Consistent with a previous study from South Korea [19], the low-vision patients in this study experienced more days of poor physical activity which did not significantly affect their mental health in relation to those without low vision. However, this effect was nullified after adjusting for age, indicating the role of age in the poor physical health reported. Other main findings are that a greater proportion of low-vision patients reported experiencing limitations in their activities, which was related to the fact that most of them had visual problems and reported more than one impairment, compared to controls. This resonates with previous studies showing poorer HRQoL [5,15,20,21,22,23] and greater risk of activity limitation [24] in adults with VI compared to controls. Low vision restricts normal visual function, which indirectly affects quality of life. Our findings show that the activity limitations faced by these individuals with low vision are related to their VI and the fact that they are more vulnerable to other impairments.

Outcomes on VRQoL also confirmed our hypothesis, since the low-vision patients had poorer VRQoL compared to controls across almost all domains. These findings are consistent with previous studies showing the negative impact of low vision on VRQoL [20,21,22,25]. Similar findings have been previously related to factors like lack of employment [22], use of avoidant coping mechanisms [21], poor emotional well-being [26], and limited access to information in low-vision patients. Although the factors predicting poor VRQoL were not explored in this study, we propose that the poor outcomes reported in our sample may be related to similar factors. Multidisciplinary rehabilitation programs have been found to improve VRQoL in patients with low vision and can be used to improve outcomes in this domain [27]. In a systematic review including 52 randomized controlled trials with 6239 participants [28], authors found that low vision rehabilitation significantly improved VRQoL, visual functioning (QoL: psychological aspect), and self-efficacy or self-esteem of participants with VI. However, the poor financial situation presents a barrier to assessing much-needed eye care services [29], especially in countries like T&T, where medical services are free in public hospitals but expensive or unaffordable in private hospitals. In most cases, people with poor financial situations will choose to join the long queue in public hospitals even in emergencies because they cannot afford the cost of private services.

In our sample, no significant difference was found in HRQoL and VRQoL outcomes among the ocular condition groups (i.e., glaucoma, retinitis pigmentosa, diabetic retinopathy, and age-related macular degeneration). This contrasts studies showing a greater risk of poor quality of life in different ocular patient groups, including patients with diabetic retinopathy [10,30,31,32]. A few factors may explain this variation in findings across studies, including differences in measures used to assess HRQoL. For example, Park et al. [19] used the European Quality of Life (EQoL) questionnaire, while this study used the CDC HRQoL-14 and the NEI-VFQ-25. A more probable explanation is the small sample sizes used in our ocular group comparison. This reduced our statistical power, since some of the comparisons emerged with moderate to large effect sizes, despite the non-significant findings. Studies involving larger sample sizes are needed to further explore HRQoL and VRQoL differences in patients with low vision in Trinidad and Tobago.

### 4.1. Strengths and Limitations of Study

There are limitations in this study that need to be considered in interpreting the results. First, the cross-sectional study design limited assessment of HRQOL-14 and NEI-VFQ-25 over time. Second, the cross-sectional nature of the study suggests that only causal inferences can be made. Third, the limited sample size, particularly for the low-vision sub-group comparisons, may have limited statistical power to detect differences in those comparisons and limits the generalizability of the findings. Fourth, the study was susceptible to recall bias, as participants might not remember the previous experience and the duration of VI was based on self-report. Fifth, the study was limited to low-vision participants with retinitis pigmentosa, glaucoma, and age macular degeneration, and findings may not be generalizable to those with other conditions. Sixth, qualitative information was not assessed from the participants. Although we did not examine the mental health aspect of people with low vision, which was shown to be higher than normal-sighted individuals [33], there are provisions for mental health management for visually impaired patients. Awareness of such services might be low among visually impaired patients; the blind welfare of T&T organizes programs to create awareness of the importance of good health, including eye and mental health. On such occasions, visually impaired patients receive opportunities to access mental health services.

Despite these limitations, the main strength of this study was the comparison of the groups (low vision and healthy individuals) using two standardized questionnaires (HRQOL-14 and NEI-VFQ-25) in data collection and methodology to investigate the quality of life in adults. Also, to our knowledge, this study is the first cross-sectional study comparing the quality of life in adults with low vision with that of a healthy group of Trinidadians. This study provides preliminary findings to support further investigations in future studies with a larger sample aimed at exploring factors predicting poor HRQOL and VRQoL in individuals with low vision.

### 4.2. Policy and Clinical Implications

There are important implications for policy and the clinical significance of the findings. The study challenges eye health professionals to implement strategies of advocacy, awareness, support, and association in addressing social determinants of low-vision patients and inequities in eye care. Therefore, it is of paramount importance to promote not only eye health, but also multidisciplinary rehabilitation programs for low-vision patients. There is a future need to raise awareness of general health, social function, and mental health among low-vision patients; integrate people-centered eye care services in clinical settings; and refer multidisciplinary teams to address population needs.

## 5. Conclusions

In summary, our results highlight the significant impact of low vision on VRQoL and HRQoL among adult Trinidadians.

A reduced quality of life was found in low-vision patients suffering from retinitis pigmentosa, diabetic retinopathy, glaucoma, and macular degeneration. This is consistent with the literature reporting on the impact of low vision on visual-related quality of life and health-related quality of life, but interestingly, we found a significantly higher number of anxious/worrying days in controls compared to the low-vision participants. In addition, there was no significant difference in HRQoL and VRQoL outcomes among the ocular condition groups (i.e., glaucoma, retinitis pigmentosa, diabetic retinopathy, and age-related macular degeneration). The information contained in this article has important clinical implications regarding offering appropriate support and interventions to improve quality of life outcomes in individuals with low vision. Future studies should include larger samples and explore factors predicting poor HRQOL and VRQoL in individuals with low vision. In addition, interventions are needed to improve the overall prognosis in this population.

## Figures and Tables

**Figure 1 ijerph-20-06436-f001:**
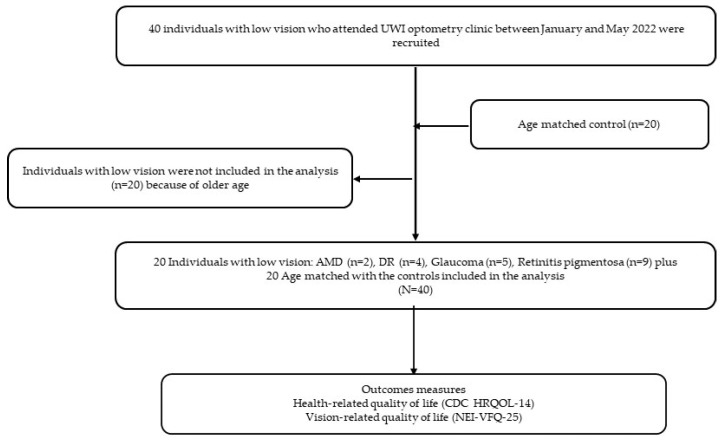
Flowchart of participant selection by ocular condition.

**Figure 2 ijerph-20-06436-f002:**
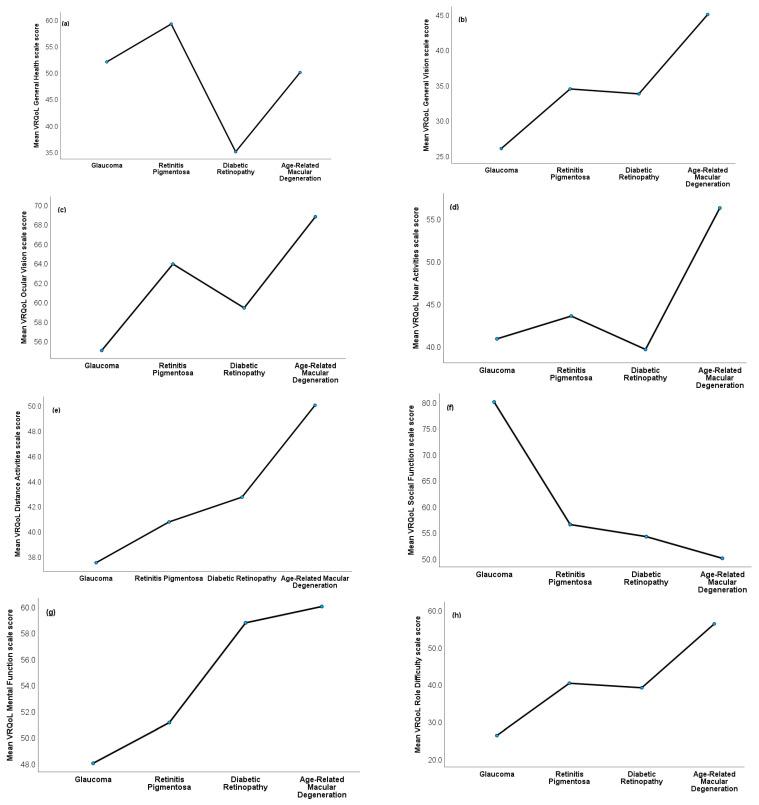
(**a**–**m**) VRQoL differences among ocular condition groups. No statistically significant differences were found among the ocular condition groups (all *p* > 0.05), but generally, patients with glaucoma reported poorer VRQOL on most scales, while patients with age-related macular degeneration reported higher VQQoL on most scales. These VRQoL scales/items of the NEI-VFQ-25 are presented: (**a**) General health scale; (**b**) general vision scale; (**c**) ocular vision scale; (**d**) near activities scale; (**e**) distance activities scale; (**f**) social function scale; (**g**) mental function scale; (**h**) role difficulty scale; (**i**) dependency scale; (**j**) driving scale; (**k**) color vision item; (**l**) peripheral vision item; and (**m**) composite score. Results based on ANOVA and χ^2^ tests.

**Table 1 ijerph-20-06436-t001:** Differences in HRQoL between low-vision (n = 20) and control (n = 20) participants.

	Unadjusted Models				Adjusted Models			
CDC HRQOL-14 Item	B	SE	95% CI	*p*	β	SE	95% CI	*p*
Healthy days core module								
Poor general health days	−1.3	0.9	−3.1, 0.4	0.133	−0.8	1.1	−3.0, 1.4	0.476
Poor physical health days	−2.8	1.2	−5.2, −0.3	0.027	−2.1	1.7	−5.4, 1.2	0.221
Poor mental health days	4.6	1.8	1.0, 8.2	0.012	1.7	2.4	−3.0, 6.4	0.481
Impact on activities days	−0.1	1.8	−2.4, 2.2	0.933	−1.6	1.6	−4.7, 1.4	0.298
Activities limitation module								
Experience activity limitation	−2.8	0.8	−4.3, −1.2	<0.001	−3.7	1.3	−6.3, −1.1	0.006
Main problems/impairment	−0.5	0.1	−0.8, −0.3	<0.001	−0.4	0.1	−0.7, −0.1	0.005
Problem duration	−4.1	2.7	−9.4, 1.3	0.140	−3.2	4.3	−11.7, 5.3	0.463
Need personal care support	−3.7	4.3	−12.1, 4.7	0.386	−0.1	0.1	−0.3, 0.0	0.125
Need support for routines	−4.1	4.3	−12.5, 4.3	0.340	−0.1	0.1	−0.3, 0.1	0.159
Healthy days symptoms module								
Days of feeling pain	9.0	5.9	−2.6, 20.6	0.129	13.3	6.9	−0.3, 26.9	0.055
Days of feeling moody	0.5	2.7	−4.9, 5.9	0.855	−2.1	3.1	−8.2, 3.9	0.491
Days of feeling anxious	13.9	4.9	4.2, 23.5	0.005	14.4	6.0	2.6, 26.1	0.017
Days of sleep problems	11.3	5.6	0.4, 22.2	0.042	11.0	6.8	−2.4, 24.3	0.109
Days of feeling energetic	−8.0	5.8	−19.4, 3.4	0.167	−8.0	7.1	−21.8, 5.9	0.260

Note: *p* < 0.05 is statistically significant. Negative β values indicate poorer HRQoL in the low-vision group compared to controls and vice-versa. Adjusted models controlled for group differences in age.

**Table 2 ijerph-20-06436-t002:** Differences in VRQoL between low-vision (n = 20) and control (n = 20) participants.

	Unadjusted Models				Adjusted Models			
NEI-VFQ-25 Subscale	B	SE	95% CI	*p*	β	SE	95% CI	*p*
General health	27.3	5.2	44.4, 58.9	<0.001	53.2	5.4	42.6, 63.8	<0.001
General vision	55.2	4.0	47.4, 63.3	<0.001	53.2	5.4	42.6, 63.8	<0.001
Ocular pain	29.4	4.0	21.5, 37.3	<0.001	24.1	5.4	13.6, 34.6	<0.001
Near activities	55.0	4.8	45.7, 64.3	<0.001	51.3	6.4	38.6, 63.9	<0.001
Distance activities	56.9	4.3	48.5, 65.2	<0.001	50.6	5.6	39.6, 61.7	<0.001
Social function	37.9	6.6	24.9, 50.9	<0.001	28.2	8.9	11.0, 45.5	0.001
Mental function	42.8	3.3	36.2, 49.3	<0.001	41.7	4.6	32.8, 50.6	<0.001
Role difficulty	61.9	3.9	54.2, 69.6	<0.001	63.0	5.4	52.5, 73.5	<0.001
Dependency	60.3	5.3	49.9, 70.7	<0.001	56.8	7.2	42.7, 70.9	<0.001
Driving	62.3	8.9	44.8, 79.8	<0.001	62.3	11.4	40.1, 84.6	<0.001
Color vision	2.1	1.5	−0.7, 5.0	0.141	1.8	1.7	-1.5, 5.2	0.280
Extreme/Moderate difficulty								
Little/No difficulty								
Peripheral vision	4.5	1.4	1.7, 7.4	0.002	3.8	1.5	0.8, 6.7	0.012
Extreme/Moderate difficulty								
Little/No difficulty								
Composite score	48.8	3.4	42.1, 55.5	<0.001	45.7	4.6	36.7, 54.7	<0.001

Note: *p* < 0.05 is statistically significant difference. Positive β values indicate better VRQoL in the control group compared to the low-vision group, and vice-versa.

**Table 3 ijerph-20-06436-t003:** Relationship between ocular conditions and HRQoL.

CDC HRQOL-14 Item	Glaucoma n (%)	Retinitis Pigmentosa n (%)	Diabetes Retinopathy n (%)	Macular Degeneration n (%)	*p*	ES
Healthy days core module						
General health days					0.592	0.309
Good to excellent	4 (80.0)	6 (66.7)	2 (50)	2 (100)		
Fair to poor	1 (20.0)	3 (30.0)	6 (60.0)	5 (50.0)		
Poor physical health days (M, SD)	5.0 (8.7)	3.6 (4.2)	3.0 (3.6)	5.0 (0.0)	0.993	0.026
Poor mental health days (M, SD)	4.0 (4.2)	1.7 (3.5)	1.3 (2.5)	0	0.471	0.142
Impact on activity days (M, SD)	1.0 (2.2)	3.0 (5.6)	3.0 (3.6)	3.5 (5.0)	0.849	0.047
Activities limitation module						
Experienced activity limitation	4 (80.0)	8 (88.9)	3 (75.0)	1 (50.0)	0.652	0.286
Main problems/Impairment					0.051	0.624
Eye/vision problem	4 (80.0)	5 (55.6)	0	0		
Two or more problems	1 (20.0)	4 (44.4)	4 (100.0)	2 (100.0)		
Problem duration (years)	5 (100.0)	9 (100.0)	4 (100.0)	2 (100.0)	-	-
Need personal care support	1 (20.0)	4 (44.4)	2 (50.0)	0	0.590	0.341
Need support for routines	2 (40.0)	7 (77.8)	2 (50.0)	0	0.185	0.491
Healthy days symptoms module						
Days of feeling pain (M, SD)	3.0 (6.7)	2.1 (3.6)	7.3 (6.1)	3.5 (5.0)	0.436	0.152
Days of feeling moody (M, SD)	5.0 (2.2)	1.7 (3.5)	1.3 (2.5)	1.5 (2.1)	0.382	0.169
Days of feeling anxious (M, SD)	2.2 (2.2)	1.7 (3.5)	4.8 (6.6)	1.5 (2.1)	0.622	0.102
Days of sleep problems (M, SD)	5.0 (7.1)	5.1 (5.2)	4.8 (6.6)	2.5 (3.5)	0.952	0.020
Days of feeling energetic (M, SD)	20.0 (10.0)	22.8 (6.7)	18.3 (9.0)	22.5 (3.5)	0.785	0.063

Note: *p* < 0.05 is statistically significant difference. Sample sizes (glaucoma = 5, retinitis pigmentosa = 9, diabetic retinopathy = 4, and age-related macular degeneration = 2). All analyses are based on χ^2^ tests, unless otherwise specified. Effect size estimation for χ^2^ tests is based on Cramer’s V values, interpreted as mild (≤0.2), moderate (0.3–0.6), and severe (>0.6). Effect size for the ANOVA models was based on eta squared values, interpreted as small effect (η^2^ = 0.01), medium effect (η^2^ = 0.06), and large effect (η^2^ = 0.14). A value of *p* < 0.05 was considered statistically significant for all analyses. No inferential statistic results were generated for the problem duration item since all participants endorsed years, with none endorsing the other options (days, weeks, months). One-way ANOVA.

## Data Availability

Data for this study are presented in the manuscript, and access to raw data can be obtained on reasonable request from the corresponding author ULO.

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
