# Peer review of "Comparative Analysis of Health- and Vision-Related Quality of Life Measures among Trinidadians with Low Vision and Normal Vision—A Cross-Sectional Matched Sample Study"

_ijerph, 2023, doi:10.3390/ijerph20146436_

Round 1

Reviewer 1 Report (Previous Reviewer 1)

The problems with the paper are as follows:

1.The conclusion is too simple, and it is necessary to summarize the main achievements and innovative points achieved in this article.

2.The outlook is relatively meaningless. It is recommended to provide practical, feasible, and meaningful methods and research directions related to the methods in this article.

3.The annotation of the horizontal and vertical coordinates in Figure 1 is unclear, so it is recommended to enlarge the text.

4.The method description of the paper suggests providing a flowchart, while strengthening the contribution and innovation points of the paper.

Minor editing of English language required

Author Response

Dear Reviewer, we have improved the relevant sections of the manuscript and provided response to your comment attached.

1) The conclusion has been revised as follows “A reduced quality of life was found in low-vision patients suffering from retinitis pigmentosa, diabetic retinopathy, glaucoma and macular degeneration. This is consistent with literature
reporting on the impact of low vision on the visual related quality of life and health-related quality of life, but interestingly, we found a significantly higher number of anxious/worrying days in controls compared to the Low vision
participants. In addition, there is no significant difference in HRQoL and VRQoL outcomes among the ocular condition
groups (i.e., glaucoma, retinitis pigmentosa, diabetic retinopathy, and age-related macular degeneration). The information contained in this article have important clinical implications regarding offering appropriate support and
interventions to improve quality of life outcomes in individuals with low vision. Future studies should include larger samples, and explore factors predicting poor HRQOL and VRQoL in individuals with low vision are needed. In addition,
interventions are needed to improve the overall prognosis in this population.”

2) The flow of the article has been revised.

3) We appreciate the suggestion. This has now been addressed in Figure 2.

4) Figure 1 has been included and shows the flowchart of participant selection and the distribution of the low vision
participants by disease status. This was also reported in the result [paragraph 1].

Language editing and grammatical errors has been done throughout the manuscript as per track changes from page 1 to page 13.

Reviewer 2 Report (New Reviewer)

The main limitation of the study is the study population. With 20 tests and 20 index participants, the authors cannot give an affirmative conclusion to the study. This provides various loopholes that can generate high bias and variance among the study population. I recommend the authors increase the study population and then do a statistical analysis, without which this study would be considered invalid

The main limitation of the study is the study population. With 20 tests and 20 index participants, the authors cannot give an affirmative conclusion to the study. This provides various loopholes that can generate high bias and variance among the study population. I recommend the authors increase the study population and then do a statistical analysis, without which this study would be considered invalid.

Author Response

Dear Reviewer,

Thanks for the very useful comments. Please find responses to the comments below. We have also revised the manuscript and all new changes are shown in red font to indicate.

Comments on the Quality of English Language

The main limitation of the study is the study population. With 20 tests and 20 index participants, the authors cannot give an affirmative conclusion to the study. This provides various loopholes that can generate high bias and variance among the study population. I recommend the authors increase the study population and then do a statistical analysis, without which this study would be considered invalid.

Response: Thanks for the comment.

Like many other studies, the small sample size is characteristic of the rare nature of the diseases especially in a small country like TnT with limited resources for early identification of low vision through population screenings. Other studies have also utilised small samples. We have acknowledged this in the limitation, however, this study provides the first and recent evidence on a vulnerable population in TnT and adds to the growing global body of evidence on the impact of visual impairment on quality of life. The relevant section in the limitation reads like this: ‘Third, the limited sample size, particularly for the low vision sub-group comparisons may have limited statistical power to detect differences in those comparisons and limits the generalizability of the findings’ [383-385].

Round 2

Reviewer 1 Report (Previous Reviewer 1)

Accept in present form.

Minor editing of English language required.

This manuscript is a resubmission of an earlier submission. The following is a list of the peer review reports and author responses from that submission.

Round 1

Reviewer 1 Report

The paper has the following problems:

1.There are only 9 references cited in the introduction, which cannot reflect the current research situation. The author is suggested to add relevant content. And add the literature of the past three years.

2.“A researcher masked to 138 the participants’ identity interviewed the participants via telephone and recorded their 139 answers.The research method is simple.

3.In line351, "Future studies should include larger sample," sample should be samples.

The workload of the paper is general and the innovation is insufficient, so it is recommended to reject it.

Author Response

1) The content has been reviewed and relevant information including literature of the past three years added (line 57 to 144). However, some of the literatures are within the past five years for instance (Jones et al., 2018).

Changes made:

  1. Vision loss has many causes (e.g., age-related macular degeneration, glaucoma), re-quires improvements in preventive and rehabilitative interventions, and often impacts on functional wellbeing and quality of life (QoL)[2], which further deteriorates with increasing severity of visual impairment[3]( . (Yibekal et al., 2020)

  1. According to the World report on vision, there are at least 2.2 billion people around the world with visual impairment, of which at least 1 billion are preventable [2]. It is estimated that blindness and vision loss resulted in 26.5 million global years of healthy life lost due to disability in 2019, and 3.1% of total global years of healthy life lost due to disability [3]. Data from 2020 shows that cataracts, glaucoma, under-corrected refractive error, age-related macular degeneration, and diabetic retinopathy are the leading causes of blindness globally in those aged 50 years and older while lead-ing causes of moderate to severe visual impairment (MSVI) were under-corrected re-fractive error and cataract [3]. In the Caribbean, a prevalence of 1.7% and 5.7% were recorded for blindness and low vision respectively [4]. It is difficult to make direct comparisons of prevalence rates in this region since data collection methods, defini-tions of visual impairment, and population characteristics vary across the region. For instance, 3.8% of Jamaicans were reported to have moderate and severe visual impair-ment in one study[5], while a prevalence rate of 2.8% was reported for visual impair-ment (moderate and severe) resulting predominantly from cataract, glaucoma, uncor-rected refractive error, diabetic retinopathy, and macular degeneration in Trinidad and Tobago (International Agency for the Prevention of Blindness), both of which are lower than the global visual impairment average of 4.5%[6]. In Trinidad and Tobago, glau-coma and cataract were also recorded as the major causes of blindness while uncor-rected refractive error and cataract were the major causes of low vision  respectively[4, 7]. These studies indicate the pervasive nature of low vision and highlight the need to understand health outcomes (e.g., QoL), especially in vulnerable individuals living with these common visual problems.

  1. The impacts of vision impairment and blindness are wide-reaching, including an increased risk of falls, cognitive impairment and dementia, depression, disability, loss of independence, and poor QoL [2, 5]. Poor visual function has been associated with poor mental health, health-related QoL (HRQoL), and vision-related quality of life (VR-QoL) in adults [8, 9] In the only study from T&T on impact of vision loss on quality of life, Braithwaite and colleagues used data from a population based national cross-sectional survey to show independent association between less severe categories of distance and near VI (NVI) and quality-adjusted life lost (QALYS) to vision impairment in adult participants aged 40 years and above [10]. According to Flitcroft et al.,( 2019)  reported relationships between low vision and health-related quality of life (HRQOL) and vision-related quality of life (VRQOL). The study  is a systematic review that aimed to evaluate the effectiveness of low vision rehabilitation for improving the quality of life of visually impaired adults. The study included 19 randomized controlled trials and found that low vision rehabilitation significantly improved both HRQOL and VRQOL in visually impaired adults (Flitcroft et al., 2019) .In an observational study on 128 persons attending a rehabilitation center for visually impaired adults in Netherlands[11], the authors reported that quality of life was more reduced among visual impaired adults than those with other chronic conditions including type 2 diabetes, coronary syndrome, and hearing impairments, but less than stroke, multiple sclerosis, chronic fatigue syndrome, major depressive disorder, and severe mental illness.

  1. The deterioration in visual function has significant impact on performing daily functions and leisure activities, leading to impaired efficiency and compromised independence of an individual [7]

  1. Visual impairment is a pervasive problem in Trinidad and Tobago [8], and such impairments have been associated with low QoL in several studies around the world[10-13]. The study conducted by Jones et al., (2018) investigated the impact of visual impairment on activities of daily living and vision-related quality of life in a sample of adults with visual impairment who are living in the UK. The study found that visual impairment had a significant negative impact on both activities of daily living and vision-related quality of life (Jones et al., 2018). Access to eye care and affordability are persistently significant issue in Trinidad and Tobago, despite the availability of numerous effective health care interventions for the prevention and control of the primary eye conditions, potentially affecting the QoL of people with vis-ual impairments in this country[14]. To date however, no observational study has in-vestigated the relationship between vision impairments and QoL in adults in Trinidad and Tobago with the only published evidence coming from a national eye health sur-vey[7] that was not specifically designed for this purpose. Such specific investigations are needed to inform clinicians, researchers, and policy makers about the impact of low vison on the livelihood of affected individuals. This can lead to the developments of further research as well as evidence-based tailored interventions and social support services aimed at improving the functional wellbeing and QoL of individuals with low vision in Trinidad and Tobago.

2) The method used in data collection has been expanded (line 189 to 202), and the number of participants have been matched with controls.

Changes made

Patients who visited the UWI Optometry Clinic during the study period and who met the eligibility criteria to participate in the study were consecutively selected using a convenient sampling method and allocated to the respective groups (low vision and control). They were invited to participate in the study through an information pamphlet which was distributed during clinic days. We obtained written and verbal con-sent from all participants and only those who consented were enrolled in the study. Prior to data collection, information regarding the study were provided to all participants via a participant information sheet and consent form. Appointments were scheduled for a telephone interview with each participant, where the CDC HRQOL-14 and NEI-VFQ-25 questionnaires were completed. All interviews were conducted by a single researcher (KE), who received extensive training on how to conduct telephone interviews with low-vision patients. Each telephone interview took approximately 30 minutes. Where a participant is unable to complete the interview within the period of appointment, another appointment was scheduled to complete the interview.

3) The sentence has been changed to samples as suggested [line 406].

Reviewer 2 Report

The article investigated the differences in health-related and vision-related quality of life measures between adults with low vision and healthy individuals in Trinidad and Tobago. It's a very important topic.

However, the text has gaps, such as the type of study, matching, number of controls and tables with excess zero values. The methodology and results need to be readjusted.

Author Response

Comments and Suggestions for Authors

The article investigated the differences in health-related and vision-related quality of life measures between adults with low vision and healthy individuals in Trinidad and Tobago. It's a very important topic. However, the text has gaps, such as the type of study, matching, number of controls and tables with excess zero values. The methodology and results need to be readjusted.

Response

The methodology and the results have been adjusted as suggested (Methodology: line 166 to 186)(Results: 287 to 340). The result has to be adjusted to align with 20 participants for each group as suggested by the reviewers.

Reviewer’s comment

Introduction:

Line 90: As a suggestion include the prevalence of visual impairment in Trinidad and Tobago.

Response

The prevalence of visual impairment in Trinidad and Tobago has been included (line 72 to 76).

Change made

while a prevalence rate of 2.8% was reported for visual impairment (moderate and se-vere) resulting predominantly from cataract, glaucoma, uncorrected refractive error, diabetic retinopathy, and macular degeneration in Trinidad and Tobago (International Agency for the Prevention of Blindness), both of which are lower than the global visual impairment average of 4.5%[6]

Reviewer’s comment

 As Trinidad and Tobago is a high-income country, is the prevalence of visual impairment lower than in other countries with a similar age and gender structure?

Response

The prevalence of visual impairment in Trinidad and Tobago is like what is obtainable in other countries with similar age and gender structure like in the Caribbean. This has now been indicated in the manuscript (line 73 to 76).

Change made

Caribbean, a prevalence of 1.7% and 5.7% were recorded for blindness and low vision  respectively [4]. It is difficult to make direct comparisons of prevalence rates in this region since data collection methods, definitions of visual impairment, and population characteristics vary across the region. For instance, 3.8% of Jamaicans were reported to have moderate and severe visual impairment in one study[5], while a prevalence rate of 2.8% was reported for visual impairment (moderate and severe) resulting predominantly from cataract, glaucoma, uncorrected refractive error, diabetic retinopathy, and macular degeneration in Trinidad and Tobago (International Agency for the Prevention of Blindness), both of which are lower than the global visual impairment average of 4.5%[6]

Reviewer’s comment

Materials and Methods:

General aspects:

  1. Was the sample defined as a convenience sample?

Response

Yes, the sample was selected consecutively selected using a convenient sampling approach. This has now been specified in the manuscript (Line 160-163)

Change made

Patients who visited the UWI Optometry Clinic during the study period and who met the eligibility criteria to participate in the study were consecutively selected using a convenient sampling method and allocated to the respective groups (low vision and control).

Reviewer’s comment

  1. Generally in case-control studies, the control number is equal to or greater than the number of cases. In the present study, it is not clear why the number of controls is smaller than the number of cases.

Response

The method has been reviewed to match the controls with the cases and we now have equal number of control (n=20) and case group (n=20) as suggested (line 144)

Changes made

Participants were 20 adults with Low vision and 20 adults with normal vision (controls).

Reviewer’s comment

  1. Explain why the authors did not use 20 cases and 20 controls and why they did not perform a matching analysis of the individuals.

Response

The study has been reviewed to use 20 cases and 20 controls as suggested with matching analysis performed. See comments above

Changes made

Participants were 20 adults with Low vision and 20 adults with normal vision (controls). The Low vision group included adults aged 18 years and over, who had previously attended the UWI Optometry Clinic (between January and May 2022), had a formal diagnosis of low vision, and were already using prescribed low vision aids. Causes of low vision in this group included glaucoma, retinitis pigmentosa, diabetic retinopathy or age-related macular degeneration.

Reviewer’s comment

  1. It is unclear what type of study. It is of the case-control type, as there is a comparison between cases and controls, and this is described in the methods.

Response

The study design has been revised to reflect case-control study as suggested (line 127 to 129).

Reviewer’s comment

2.2 Eligibility criteria and participants

Case: Aged 18 years and over, who attended the BLINDED FOR REVIEW optometry clinic between January and May 2022, and were already using the prescribed low vision aids.

Controls: The Control group included adults with healthy eyes, who were free from ocular disease or any visual impairment upon clinical evaluation. Was the control group recruited from the same location (optometry clinic)? (a) Persons aged 18 years or older, without diabetic retinopathy, retinitis pigmentosa, glaucoma and macular degeneration being treated in the outpatient clinic? Or (b) recruits from the same household; or (c) family members without the morbidity; or (d) peoples in the same community?

Please clarify if the controls are hospital or community and how they were recruited.

Response

The controls were from community members and this has been clarified in the methods Line 149 to 153.

Changes made

Participants were 20 adults with Low vision and 20 adults with normal vision (controls). The low vision group included adults aged 18 years and over, who had previously attended the UWI Optometry Clinic (between January and May 2022), had a formal diagnosis of low vision, and were already using prescribed low vision aids. Causes of low vision in this group included glaucoma, retinitis pigmentosa, diabetic retinopathy or age-related macular degeneration. The control group was made up of adult participants (staff and students) from the university community, without diabetic retinopathy, retinitis pigmentosa, glaucoma and macular degeneration, and who had a best corrected visual acuity of 20/20. This group underwent comprehensive eye tests (e.g., visual acuity, intraocular pressures, retinal examination and slit lamp examination) to rule out any ocular problems.

Exclusion criteria for both groups (i.e., Low vision and controls) were: (1) individuals under 18 years old, (2) individuals with corneal defects or anterior surface eye diseases such as keratoconus, Fuchs' endothelial dystrophy, and bullous keratopathy, and (3) individuals with cognitive impairments such as Alzheimer's or dementia as recorded in their hospital records.

Reviewer 3 Report

This research brings additional information about the impact of visual impairments on the studied population. The study investigated differences in health- and vision-related QoL measures between adults with low vision and healthy individuals in Trinidad.

  In order to improve the quality of the publication, I am asking the authors for the following clarifications:

1. How does good financial situation/political stability correlate with inequity regarding access to eye health medical services?

2. Is there a correlation with poor health education in the studied area, regarding the importance of eye health on global health? Are there such projects carried out by the health authorities? How about the mental health management in visually impaired patients?

Author Response

2.5 Statistical analysis

Lines 179 – 180: Demographic differences in age and sex were evaluated using independent 179 sample t-test and Fishers exact test, respectively...

Please, rewrite the text, as differences by sex (categorical variable) do not apply to Student's t test.

Response

instead as a continuous variable, as indicated by the mean and standard deviation values reported in the results (line359 to 360): with the low vision group reporting a higher average age (mean = 61.6, SD = 16.0) compared to the Control group (mean = 29.0, SD = 9.8). Hence, this statement has been maintained by the authors.

Reviewer’s comment

Results:

As the sample obtained is small, as a suggestion, use only the total score of the instruments.

Response

We thank the reviewer for this observation and agree that the sample sizes in the subgroups of this study is indeed small. We have hence decided …The total score of the instrument has been used.

Reviewer’s comment

There are many items with zero value in the tables but with calculated statistics and this needs to be fixed in the analyses. Once again, as the sample is relatively small, as a suggestion, do not use too many categories in the variables so as not to have too many zero values.

As suggested, use matching data to control potential confounding.

Response

There are only 5 categories with no participants in the table. However, the statistical values provided in relation to these values excluded these category in the analysis. The data were coded in this format (e.g., number of days of depression on the HRQOL) to reflect the state of the sample, and the authors would like to maintain that.

Reviewer 3

Comments and Suggestions for Authors

This research brings additional information about the impact of visual impairments on the studied population. The study investigated differences in health- and vision-related QoL measures between adults with low vision and healthy individuals in Trinidad.

  In order to improve the quality of the publication, I am asking the authors for the following clarifications:

  1. How does good financial situation/political stability correlate with inequity regarding access to eye health medical services?

Response

This is a good point and we have incorporated that in our discussion. The section now reads:

However, poor financial situation presents a barrier to assessing the much needed eye care services [29] especially in countries like T&T  where medical services are free in public hospitals but expensive or unaffordable in private hospitals. In most cases, people with poor financial situation will choose to join the long que in public hospitals even in emergency because they cannot afford cost for private services

  1. Is there a correlation with poor health education in the studied area, regarding the importance of eye health on global health?

Response

Overall, there is a good awareness of the importance of health education in the area although there is no literature to back it up, but other factors could also influence or deter people from taking good care of their health thus reflecting as poor health education. No change was made.

Reviewer’s comment

Are there such projects carried out by the health authorities?

Response

 There were few times when such projects were carried out by different non-governmental organizations but not so often do they embark on such project to educate the public on the importance of eye health, and no evidence to show the impact or evaluation. A lot still needs to be done in that regard.

Reviewer’s comment

How about the mental health management in visually impaired patients?

Response

Thanks for the suggestion. We have included a discussion about mental health in the manuscript. The section now reads:

Although we did not examine the mental health aspect of people with low vision which were shown to be higher than normal sighted individuals[33], there are provisions for mental health management for visually impaired patients. Awareness of such services might be low among visually impaired patients; the blind welfare of T&T organizes programs to create awareness on the importance of good health including eye and mental health. On Such occasions visually impaired patients get opportunities to access mental health services.